# *Salmonella* Typhimurium and *Pseudomonas aeruginosa* Respond Differently to the Fe Chelator Deferiprone and to Some Novel Deferiprone Derivatives

**DOI:** 10.3390/ijms221910217

**Published:** 2021-09-23

**Authors:** Serena Ammendola, Valerio Secli, Francesca Pacello, Martina Bortolami, Fabiana Pandolfi, Antonella Messore, Roberto Di Santo, Luigi Scipione, Andrea Battistoni

**Affiliations:** 1Department of Biology, University of Rome “Tor Vergata”, Via della Ricerca Scientifica, 00133 Rome, Italy; serena.ammendola@uniroma2.it (S.A.); valsecli@outlook.it (V.S.); francesca.pacello@uniroma2.it (F.P.); 2Department of Basic and Applied Sciences for Engineering, Sapienza University of Rome, Via del Castro Laurenziano 7, 00161 Rome, Italy; martina.bortolami@uniroma1.it (M.B.); fabiana.pandolfi@uniroma1.it (F.P.); 3Department of Chemistry and Drug Technology, Sapienza University of Rome, Piazzale Aldo Moro 5, 00185 Rome, Italy; antonella.messore@gmail.com (A.M.); luigi.scipione@uniroma1.it (L.S.); 4Istituto Pasteur, Fondazione Cenci Bolognetti, Department of Chemistry and Technology of Drug, Sapienza University of Rome, Piazzale Aldo Moro 5, 00185 Rome, Italy; roberto.disanto@uniroma1.it

**Keywords:** iron transport, chelating agents, antimicrobials, *Pseudomonas aeruginosa*, *Salmonella* Typhimurium

## Abstract

The ability to obtain Fe is critical for pathogens to multiply in their host. For this reason, there is significant interest in the identification of compounds that might interfere with Fe management in bacteria. Here we have tested the response of two Gram-negative pathogens, *Salmonella enterica* serovar Typhimurium (STM) and *Pseudomonas aeruginosa* (PAO1), to deferiprone (DFP), a chelating agent already in use for the treatment of thalassemia, and to some DFP derivatives designed to increase its lipophilicity. Our results indicate that DFP effectively inhibits the growth of PAO1, but not STM. Similarly, Fe-dependent genes of the two microorganisms respond differently to this agent. DFP is, however, capable of inhibiting an STM strain unable to synthesize enterochelin, while its effect on PAO1 is not related to the capability to produce siderophores. Using a fluorescent derivative of DFP we have shown that this chelator can penetrate very quickly into PAO1, but not into STM, suggesting that a selective receptor exists in *Pseudomonas*. Some of the tested derivatives have shown a greater ability to interfere with Fe homeostasis in STM compared to DFP, whereas most, although not all, were less active than DFP against PAO1, possibly due to interference of the added chemical tails with the receptor-mediated recognition process. The results reported in this work indicate that DFP can have different effects on distinct microorganisms, but that it is possible to obtain derivatives with a broader antimicrobial action.

## 1. Introduction

Iron (Fe) is the fourth major element on the Earth’s crust and the most abundant transition metal in most organisms, where it is a cofactor in many enzymes dealing with cellular respiration, DNA synthesis and repair and response to oxidative stress [1,2]. Despite its abundance in the environment, Fe is not easily absorbed by organisms. In fact, while under anaerobic conditions and low pH the dominant oxidation state of Fe is the highly soluble ferrous ion (Fe^2+^), and in environments characterized by aerobic conditions and neutral pH it is mainly found as poorly soluble ferric ions (Fe^3+^). Moreover, Fe uptake and its homeostasis must be strictly controlled because an excess of this metal can induce the formation of reactive oxygen species through the Haber–Weiss and Fenton reactions, that can lead to severe damages in lipids, proteins and DNA [3].

The problem of an adequate Fe recruitment is extremely critical for pathogenic bacteria, as they colonize environments where this metal is scarcely available in accessible forms. In fact, the host nutritional immunity response involves various strategies aimed at the sequestration of transition metals to starve microorganisms from nutrients which are essential for their growth, such as the reduction of Fe concentration in plasma and the secretion of Fe-sequestering molecules [4]. Given this, the ability of a microorganism to recruit Fe in the host tissues is considered one of the most important factors that determines the proficiency of pathogens to multiply in the host and cause the disease [5]. Thus, to ensure an adequate supply of Fe in the most diverse environmental situations, microorganisms have developed different systems of Fe import.

In anaerobic or microaerobic conditions, Gram-negative bacteria import Fe^2+^ using a series of inner membrane transporters, as this ion is believed to enter the periplasm through the porins of the outer membrane. Some of these importers have a broad metal specificity and are involved in the uptake of other divalent cations, such as manganese (Mn^2+^) and zinc (Zn^2+^). The Fe^2+^ uptake system most widely distributed among bacteria is the Feo system, which is also considered a virulence factor for some pathogenic species [6,7,8,9].

In aerobic conditions, the import of the poorly soluble Fe^3+^ is mainly achieved through the secretion of siderophores, low-molecular-weight molecules that capture the metal in the extracellular environment and allow its internalization through specific outer membrane receptors. Siderophores are all characterized by a very high Fe-binding ability and are usually classified in four major different chemical classes: cathecolates (e.g., enterobactin, vibriobactin), phenolates (e.g., pyochelin, yersiniabactin), hydroxamates (e.g., alcaligin, rhequichelin) and carboxylates (e.g., staphyloferrinA, rhizoferrin) [10]. Siderophores confer to bacteria the ability to colonize Fe-restricted environments, including host tissues. In fact, their high affinity for Fe^3+^ makes them highly competitive against host Fe sequestration strategies, as demonstrated by the attenuation of pathogens impaired in the synthesis or in the acquisition of siderophores [11,12,13,14,15,16]. The importance of siderophores for the ability of bacteria to obtain Fe in different environmental niches is furtherly underlined by the presence of apparently redundant systems in the same microorganism and by the ability of many bacteria to exploit siderophores produced by other species, thus increasing their fitness in the colonization of Fe-restricted environments.

Considering the importance of Fe acquisition processes for the ability of pathogens to cause disease, it is not surprising that many studies have focused on the possibility of controlling microbial infections through the use of chelating agents (for a recent review, see [17]). However, the results of these studies are controversial, both for the possible toxicity of chelators and for an evident variability of the effects of these agents on different pathogens, including the potential capability of some chelators to act as Fe transporters themselves [18].

Among Fe chelators, 3-hydroxy-1,2-dimethyl-4(1*H*)-pyridone (deferiprone, DFP) is a synthetic drug successfully used in the treatment of thalassemic patients with Fe overload [19]. The use of DFP has also been proposed for the treatment of other pathologies related to Fe accumulation and ROS generation, such as some neurodegenerative diseases [20]. Interestingly, DFP has also been shown to have antibacterial and antifungal activity on some common nosocomial infectious agents, such as clinical strains of *Pseudomonas aeruginosa*, *Yersinia enterocolitica*, *Staphylococcus aureus* and *Vibrio vulnificus* [21,22,23,24]. Furthermore, DFP has been reported to impair biofilm formation [25,26]. The FDA approval of the use of DFP for the treatment of human diseases and its ability to inhibit microbial growth in vitro suggest that this molecule may be a particularly promising chelator for use in antimicrobial therapies. Due to its small dimension and neutral charge, DFP is thought to easily cross the biological membranes, probably including the bacterial ones. It has been hypothesized that the Fe sequestering ability of DFP can be exploited both in the extracellular environment and inside the bacterial cells [22], however evidences about its mechanism of action is still lacking.

Pursuing our previous studies on potential antimicrobial drugs interfering with Fe in free form [27] or coordinated in porfirines [28]. We decided to better understand the antimicrobial potential of DFP and conducted studies on the effects of this compound on two different pathogenic microorganisms, *Salmonella enterica* sv Typhimurium and *Pseudomonas aeruginosa*. Further studies were conducted on a new DFP fluorescent derivative, to evaluate the capability of DFP to permeate the membranes of these microorganisms, and on some DFP derivatives, already studied for their activity against *Candida albicans* planktonic cells and biofilm [27].

These compounds have been designed to increase lipophilia of DFP, by connecting it with different aryl-alkyl groups or with some non-steroidal anti-inflammatory drugs (NSAIDs). The results reported in this study show that DFP and its derivatives have very different effects on the two microorganisms, which can be traced back to the ways in which these molecules penetrate inside the bacterial cells.

## 2. Results

### 2.1. Salmonella Typhimurium and Pseudomonas aeruginosa Show Different Susceptibility to DFP

We started this study by evaluating the effect of DFP on the growth of two human pathogens, *Salmonella enterica* serovar Typhimurium (reference strain ATCC^®^ 14028^™^, hereafter referred to as STM) and *Pseudomonas aeruginosa* (reference strain PAO1, hereafter referred to as PAO1). To put results in relation with the ability of these bacteria to efficiently acquire Fe from the culture medium, we have compared the growth of wild type STM with an isogenic *fepA/entF* mutant [29]. It is unable to acquire Fe through enterobactin, the major STM siderophore, and the growth of wild type PAO1 with a mutant strain lacking the *pchD* and *pvdA* genes, and is therefore unable to produce the two *P. aeruginosa* siderophores pyochelin and pyoverdine (Appendix A).

The results shown in Figure 1 indicate that the effect of DFP is dependent on the bacterial species and can be rescued by Fe supplementation. DFP has a negligible effect on the growth of wild type STM, that can be appreciated only at the higher concentration tested (2 mM). However, the compound shows a significant ability to inhibit the growth of the *fepA/entF* mutant (Figure 1a). In comparison, PAO1 shows a more pronounced sensitivity to DFP, with a decreased growth rate even at the lowest concentration tested (0.5 mM) and a complete inhibition at 2 mM. There are no substantial differences in the DFP-mediated growth inhibition between wild type PAO1 and the mutant strain unable to produce pyochelin and pyoverdine (Figure 1c). In both bacterial species, the growth inhibition related to the presence of DFP can be rescued by Fe supplementation (Figure 1b,d), either in the wild type strains or in the mutants, though not completely in the *fepA/entF* strain.

These results suggest that in the case of *S.* Typhimurium, DFP competes with enterochelin for the binding of extracellular Fe, while the chelator does not appear to compete with the siderophores of *P. aeruginosa*.

### 2.2. DFP Differently Modulates the Expression of Fe-Dependent Genes in P. aeruginosa and in S. Typhimurium

To confirm the ability of DFP to interfere with Fe homeostasis, we evaluated its effects on some Fur-regulated elements. To this aim, we have chosen to analyze the accumulation of the STM proteins IroB (a glycosyl transferase involved in the synthesis of the siderophore salmochelin) and SodB (a cytoplasmic Fe cofactored superoxide dismutase) and the activity of the PAO1 promoters *pchR* (a regulator of the ferripyochelin receptor gene) and *pvdS* (an alternative sigma factor that controls pyoverdine biosynthesis).

IroB and SodB accumulation was analyzed by Western Blots on two STM strains that carry an epitope tagged version of the proteins [29,30]. This was found to be strictly dependent on Fe availability, as shown in Figure 2a. In fact, IroB is repressed as Fe concentration increases, while SodB is absent in low Fe conditions and accumulates under high Fe availability. When bacteria are cultivated in Fe-rich medium (LB), the Fe chelator 2,2′-Bipyridine (Bpy) induces IroB accumulation, whereas DFP has no detectable effect. In the same medium, SodB production is abolished by Bpy and is only slightly downregulated by DFP.

In contrast, in PAO1 the transcriptional activity of the two Fur-regulated promoters *pchR* and *pvdS* was similarly modulated by DFP, the latter being even more induced with DFP than with Bpy treatment (Figure 2b). As Bpy is a membrane permeable chelator, these results support the hypothesis that DFP can easily enter within PAO1, whereas its effects on STM are more compatible with an extracellular sequestration ability. The ability of DFP to induce iron deficiency in PAO1 was also confirmed by the increase in production of pyoverdine (Appendix A).

### 2.3. DFP Can Easily Penetrate in P. aeruginosa, but Not in S. Typhimurium

To verify the hypothesis that DFP has a different capability to permeate the membranes of STM and PAO1 we have synthesized compound **1**, where DFP is conjugated to the fluorophore 4-amino-7-nitrobenzofurazan (Figure 3a). This compound has an absorption peak at 480 nm (Figure 3b) and therefore it could be imaged under a fluorescence microscope using the FITC excitation filter. The microscopic analysis of PAO1 and STM grown for a short time (20 min) in presence of compound **1** revealed that, while in the STM no fluorescence signal was associated to the cells, PAO1 strongly fluoresced (Figure 3c). These results confirm that DFP interaction with bacteria depends on the species considered and support the hypothesis that DFP can permeate more easily PAO1 than STM. Interestingly, PAO1 incubated with the fluorophore not conjugated to DFP was not fluorescent (Appendix A), indicating that the DFP moiety has a critical role in membrane permeation.

### 2.4. Antimicrobial Activity of DFP Derivatives

To further investigate the ability of DFP to interfere with metal homeostasis in bacteria, we have screened a collection of DFP derivatives, all characterized by the addition of lipophilic tails (Appendix A), by analyzing their effect on the growth of STM, compared to Bpy or DFP treatment. Interestingly, all the compound listed in Appendix A with the DFP-hydroxyl function blocked with the benzyl group (indicated with the letter **a** immediately after the number) proved to be completely unable to inhibit bacterial growth or to affect Fe homeostasis, indicating that the biological effects of these derivatives are essentially related to the iron-binding ability of the DFP moiety. Figure 4 focuses on a subset of the tested compounds, including the three causing the most important impairment of STM growth (2b, 3b and 7b). To show the critical importance of the hydroxyl function of DFP, this figure includes also compound 2a. The structures of these compounds are reported in Figure 4a.

As shown in Figure 4b, we have observed that the compounds 2b and 3b significantly reduce growth of wild type STM, thereby showing an enhanced effect compared to DFP. Compound 7b has an activity comparable to that of DFP, while treatment with compound 2a has no effect on STM growth. All the compounds, with the exception of 2a, had a more pronounced inhibitory effect on the *fepA/entF* strain than on wild type STM.

In PAO1 (Figure 4c), compound 3b shows an inhibitory activity comparable to that of DFP, compound 7b has a slightly lower activity, whereas compound 2b is not inhibitory at all. As in the case of STM, compound 2a has no effect on PAO1 growth. Noticeably, compounds 2b and 7b show a higher inhibitory activity on the *pchDpvdA* mutant strain than on the wild-type strain.

Based on the results from this screening, we have chosen compounds 2b and 3b for further analyses.

### 2.5. Detailed Analysis of Selected Compounds

We have analyzed the response of STM to different doses of compound 3b and the effect of Fe supplementation in restoring growth. As shown in Figure 5a, compound 3b is capable to slow the growth of wild type STM starting from 0.5 mM and the effect is enhanced as the dose increases. This growth phenotype is almost completely rescued by Fe supplementation in the wild-type strain. Also, in the case of the *fepA/entF* mutant strain, compound 3b shows high activity and a significant increase in growth can be observed in the presence of Fe (Figure 5c). We have also analyzed the effect on STM of compound 3b in a chemically defined medium with low Fe availability (M9 minimal medium), in order to compare its activity with that of other Fe chelators, namely DFP and Bpy. As depicted in Figure 5d, this compound has an effect comparable to that of Bpy and the impairment in enterobactin production causes a greater inhibition of *fepA/entF* growth. Moreover, the induction of IroB protein accumulation, shown in Figure 5b, suggests that in presence of compound 3b, STM senses an Fe starvation. Similar results were obtained for compound 2b (data not shown). As the modifications introduced in DFP are not expected to significantly change the affinity of the compound for Fe, these observations suggest that DFP derivatives with an enhanced hydrophobic profile could have an enhanced ability to permeate bacterial membranes and to interfere with intracellular Fe management.

The induction of an Fe starving response by compound 3b also emerged from the analysis of the transcriptional activity of three Fur-dependent promoters in PAO1. In fact, as reported in Figure 6, the promoters of *pchR*, *pvdS* and *feoA* are all positively regulated when the strains are grown in a medium supplemented with compound 3b, as well as with Bpy. Moreover, these inductions are always reversed by the addition of Fe in the culture medium.

## 3. Discussion

In the first part of this study, we compared the effect of DFP on the growth of two different Gram-negative pathogens, observing very different results. Indeed, a 2 mM concentration of DFP completely inhibits the growth of PAO1, but very modestly interferes with the growth of wild-type STM. In the case of PAO1, no significant differences are observed in the sensitivity to DFP between the wild strain and a mutant strain unable to synthesize the two siderophores pyochelin and pyoverdine. Conversely, the growth of an STM mutant unable to synthesize the siderophore enterochelin is inhibited by DFP, although not at the level shown by PAO1. These results could be explained by hypothesizing that in *P. aeruginosa* DFP is easily internalized by the microorganism, interfering with the intracellular management of Fe, and that DFP has a lower ability to penetrate STM, thus competing with enterochelin for the binding of extracellular Fe. This hypothesis is in line with the observation that in PAO1 DFP induces the expression of two Fe-regulated genes, *pchR* and *pvdS*, comparably to the membrane-permeant chelator Bpy, which interferes with the intracellular pool of Fe^2+^. In contrast, in STM a high concentration of DFP has a very low effect on the intracellular accumulation of IroB and SodB, two proteins that are highly responsive to Bpy.

To prove the hypothesis of a differential capability of DFP to cross the membranes of STM and PAO1, we synthesized a fluorescent derivative of DFP and tested its uptake in bacteria. The results reported in Figure 3 indicate that after a short incubation period (20 min) in presence of this compound PAO1 becomes fluorescent, with a distribution of the fluorescence suggestive of a preferential accumulation of the compound in the periplasmic space. In contrast, no fluorescence signal was observed in STM under the same incubation conditions. Taken together, these results indicate that DFP penetrates PAO1 with great ease, but not STM. Although additional studies are necessary to characterize the route of DFP entry, this result suggests that the outer membrane of PAO1 could contain a channel that allows DFP entry and that a similar channel is not present in STM.

To further investigate the possibility of using the Fe sequestration capacity of DFP in an antibacterial function, we decided to test a series of derivatives of the chelator characterized by a greater lipophilic character. The synthesis of these derivatives required the protection of the hydroxyl function of DFP with a benzyl group. All the benzyl-containing compounds, with reduced ability to bind Fe, did not show any antibacterial activity, thus confirming that the antimicrobial effects of these compounds are related to interference with Fe homeostasis. Some of the tested derivatives (compounds 2b and 3b) exhibited enhanced antimicrobial activity towards either wild type and mutant STM, with compound 3b being the most active. This compound has an inhibitory effect comparable to that of Bpy and induces the intracellular accumulation of the Fe-responsive protein IroB. These observations suggest that compound 3b penetrates much more easily into STM than DFP, probably due to its greater lipophilic character. The activity of these same compounds on PAO1 was instead variable. Compound 3b proved to be as active as DFP, compound 7b was slightly less active, whereas compound 2b had no activity. Based on the hypothesis that DFP quickly penetrates PAO1 through a selective receptor, it is possible to suggest that the introduction of an aryl group into compound 3b does not alter the recognition of the receptor and does not modify the input kinetics of the compound, which would instead be slowed down by the introduction of a more complex chemical group, as in compound 7b. The complete loss of activity of compound 2b, which differs from compound 3b just for the lack of a methylene group, was more unexpected and suggests that even subtle changes in the nature of the tail added to the DFP moiety can modulate the receptor-mediated recognition process or the productive release of the chelating agent inside the cell.

The results reported in this study indicate that DFP can have distinct biological effects on different microorganisms, and that therefore the potential use of DFP as an antimicrobial must be carefully evaluated according to the target species. This may depend on factors such as the different ability of the compound to cross membranes and the ability of different microorganisms to produce siderophores with an affinity for Fe higher than that of DFP. For example, it is known that the enterochelin produced by STM has an exceptionally high affinity for Fe^3+^ (10^43^ M^−1^) [31], largely higher than that of DFP (10^35^ M^−1^) [32]. It is therefore unable to compete efficiently for the binding of extracellular Fe. The *Pseudomonas* siderophores pyoverdine and pyochelin have lower Fe affinity constant (10^32^ and 10^18^ M^−2^, respectively) [33,34]. With respect to enterochelin, this is not sufficient to explain the very strong effect of DFP on PAO1. In this case, the strong inhibitory effect of the chelating agent seems to correlate with an increased ability to permeate the bacterial membranes, strongly suggestive of the presence of an outer membrane protein able to favor the entry of DFP. Future studies will be addressed to the identification of the possible receptor involved in DFP uptake, but it is worth nothing that the structure of DFP resembles the structure of some natural siderophores, such as cepaciabactin [35]. Since it is known that *Pseudomonas* is capable of internalizing siderophores produced by other microbial species [36]. It is possible to hypothesize that DFP can enter through channels used for the acquisition of these molecules.

Another interesting aspect that emerges from this study is that some of the derivatives tested in this study turned out to be much more active than DFP against STM, probably because the modifications introduced allow the chelator to more easily penetrate the cell due to an increase in the lipophilicity of the compound. Particularly interesting is the compound 3b, which proved to be highly active both against STM and PAO1. The same compound had already shown itself to be highly active against *Candida albicans* as well as devoid of toxicity against *Galleria mellonella* larvae [27]. The combination of these observations identifies 3b as a compound with a much broader spectrum of action than DFP.

In conclusion, our study suggests that the antimicrobial activity of DFP can be largely modified through the introduction of limited chemical modifications in its structure.

## 4. Materials and Methods

### 4.1. Bacterial Strains, Media and Chemicals

*Salmonella enterica* ser. Typhimurium and *Pseudomonas aeruginosa* strains used in this study are listed in Appendix A. Bacteria were routinely grown in Luria-Bertani (LB) medium (Bacto tryptone 10 g L^−1^, yeast extract 5 g L^−1^, NaCl 10 g L^−1^), at 37 °C with constant aeration. Fe-limiting conditions were achieved employing M9 Minimal Medium (NaCl 0.1 g L^−1^, NH_4_Cl 0.1 g L^−1^, Na_2_HPO_4_ 2H_2_O 1.02 g L^−1^, KH_2_PO_4_ 0.6 g L^−1^, MgSO_4_ 1 mM, CaCl_2_ 0.1 mM and 0.2% (*w*/*v*) glucose, pH 7.2) or VBMM as previously described [37]. More stringent Fe needs were achieved by using the M9- Succinate medium (M9-S), where 0.45 % (*w*/*v*) succinate is present instead of glucose as the carbon source.

All antibiotics were provided by Sigma-Aldrich (Milano, Italy), sterilized by filtration and stored at −20 °C, were used at the following concentrations: for *E. coli*, 50 mg L^−1^ kanamycin and 10 mg L^−1^ tetracycline; for *P. aeruginosa*, 100 mg L^−1^ tetracycline. 2,2’-Bipyridine (Bpy) was dissolved in DMSO and FeSO_4_ (Sigma-Aldrich, Milano, Italy) was dissolved in ultra-pure water, both as 50 mM stock solutions.

### 4.2. DFP, Derivatives and Fluorescent Compounds

All reagents, solvents and deuterated were of high analytical grade and were purchased from Sigma-Aldrich (Milano, Italy). ^1^H and ^13^C NMR spectra were recorded on AVANCE-400 Bruker spectrometer (9.4 T) operating at 400 and 100 MHz, respectively; chemical shifts (δ) are given in ppm, relatively to TMS; coupling constant are given in Hz. The following abbreviation were used: s = singlet, d = doublet, t = triplet, m = multiplet, bs = broad singlet, bm = broad multiplet. Melting points were determined on FALC Mod. 360 D apparatus and are uncorrected. Mass spectra were recorded on a ThermoFinnigan LCQ Classic LC/MS/MS ion trap equipped with an ESI source and a syringe pump. Samples (10^−4^–10^−5^ M in MeOH/H_2_O 80:20) were infused in the electrospray system at a flow rate of 5–10 µL min^−1^.

The 3-Hydroxy-1,2-dimethyl-4(1*H*)-pyridone (DFP) was purchased from Sigma-Aldrich (Milano, Italy), the compounds 2a,b–8a,b were synthesized as previously reported [27]. The compound 1 was synthesized as indicated below.

*Synthesis of the intermediate 3-(benzyloxy)-2-methyl-1-(4-((7-nitrobenzo[c][1,2,5]oxadiazol-4-yl)amino)butyl)pyridin-4(1H)-one.* 1-(4-aminobutyl)-3-(benzyloxy)-2-methylpyridin-4(1H)-one (0.143 g, 0.5 mmol) was done as previously reported [38]. This was dissolved in 5.5 mL of MeOH. To this solution, placed in an ice bath at 0 °C, a solution of 4-chloro-7-nitrobenzofurazan (0.100 g, 0.5 mmol) and N,N-diisopropylethylamine (DIPEA, 94 µL, d = 0.755 g/mL, 0.55 mmol) in 5.5 mL of MeOH was added dropwise in about 20 min. The obtained mixture was stirred at room temperature under inert atmosphere for 22 h. Then the solvent was removed under reduced pressure; the residue was diluted in 50 mL of CH_2_Cl_2_ and washed with a saturated aqueous solution of Na_2_CO_3_ (3 × 50 mL). The organic layer was dried over Na_2_SO_4_, filtered and concentrated under reduced pressure. The crude material was purified by column chromatography on silica gel (CH_2_Cl_2_/MeOH/TEA 9:1:1, Rf = 0.65) and subsequently the product was crystallized from acetone with n-hexane, to afford the desired compound. Orange solid, 0.136 g, 61% yield. ^1^H NMR (400 MHz) (DMSO-d_6_) δ (ppm): 9.53 (1H, t, J = 5.44 Hz, -NH-) 8.51 (1H, d, J = 8.96 Hz, benzofurazan); 7.61 (1H, d, J = 7.52 Hz, -N-CH=CH-C=O); 7.40–7.29 (5H, m, aromatic); 6.42 (1H, d, J = 8.96 Hz, benzofurazan); 6.15 (1H, d, J = 7.52 Hz, -N-CH=CH-C=O); 5.01 (2H, s, Ar-CH_2_-O-); 3.91 (2H, t, J = 6.76 Hz, -CH_2_-N-); 3.49 (2H, bm, -NH-CH_2_-); 2.17 (3H, s, -CH_3_); 1.72-1.63 (4H, m, -NH-CH_2_-CH_2_-CH_2_-CH_2_-N-).

*Synthesis of 3-hydroxy-2-methyl-1-(4-((7-nitrobenzo[c][1,2,5]oxadiazol-4-yl)amino)butyl)pyridin-4(1H)-one (1).* The intermediate 3-(benzyloxy)-2-methyl-1-(4-((7-nitrobenzo[*c*][1,2,5]oxadiazol-4-yl)amino)butyl)pyridin-4(1*H*)-one (0.136 g, 0.30 mmol) was dissolved in 5 mL of aqueous 6 M HCl and the solution was heated to reflux for 6 h. Then, the reaction mixture was concentrated in vacuum and the residue was diluted with aqueous 2 M Na_2_CO_3_ until pH = 10. The mixture was evaporated in vacuum and then washed with warm CH_3_CN (5 × 10 mL) and H_2_O (3 × 1 mL). The residue was dried in vacuum with P_2_O_5_, to afford compound **1**. Dark red solid, 0.020 g, 19% yield. m.p. = 216–218 °C ^1^H NMR (400 MHz) (DMSO-*d_6_*) δ (ppm): 9.53 (1H, bs, -NH-) 8.50 (1H, d, *J* = 8.72 Hz, benzofurazan); 7.57 (1H, d, *J* = 7.28 Hz, -N-CH=CH-C=O); 6.42 (1H, d, *J* = 9.00 Hz, benzofurazan); 6.10 (1H, d, *J* = 7.24 Hz, -N-CH=CH-C=O); 3.97 (2H, t, *J* = 6.72 Hz, -CH_2_-N-); 3.45 (2H, bm, -NH-CH_2_-); 2.29 (3H, s, -CH_3_); 1.79-1.64 (4H, m, -NH-CH_2_-CH_2_-CH_2_-CH_2_-N-). ^13^C NMR (100 MHz) (DMSO-*d_6_*) δ (ppm): 168.9; 145.5; 145.2; 144.5; 144.2; 137.9; 137.6; 128.5; 120.6; 110.5; 99.2; 52.4; 42.9; 27.7; 24.6; 11.4. ESI–MS (m/z): [M–H]^−^ = 358.9.

### 4.3. Analyses of Bacterial Growth

Single colonies were pre-inoculated for five hours in LB, then diluted 1:500 in fresh medium supplemented or not with the indicated treatments. A volume of 0.2 mL of each sample was inoculated in a 96-microwell (Greiner Bio-One, Austria), incubated at 37 °C in a Sunrise^™^ microplate reader (Tecan, Männedorf, Switzerland) and Optical density at 595 nm (OD_595_) has been registered every hour for 16 h. Each sample was tested in triplicate.

### 4.4. pchR, pvdS and feoA Promoters’ Activity Assay

The promoter regions of *pchR*, *pvdS* and *feoA* genes, carried on the pMP220 reporter plasmid, were mobilized from *E. coli* DH5α into PAO1 strain by triparental mating, using *E. coli* HB101 pRK2013 as the helper strain (Appendix A) with standard procedures already described [39]. The PAO1 exconjugants carrying reporter plasmids prom-*pchR* pMP220, prom-*pvdS* pMP220 and prom-*feoA* pMP220 were grown over-night in LB medium and assayed for their b-galactosidase activity as previously described [40]. For each condition three independent inoculates were tested and absorbances were recorded in triplicate in a Sunrise^™^ microplate reader (Tecan, Männedorf, Switzerland).

### 4.5. SDS-PAGE, Western Blotting and Immunodetection

Equal amounts of overnight grown bacteria (approximately 2.5 × 10^8^ cfu/sample), normalized according to their optical density at 600 nm (Lambda9 spectrophotometer, PerkinElmer), were lysed in Sample Buffer and protein contents were denatured for 8 min at 95 °C, loaded on a 12% SDS-PAGE and blotted onto nitrocellulose membranes (Hybond ECL; GE-Healthcare, Chicago, Illinois, USA), following standard procedures already described [41]. Epitope-tagged proteins were immunodetected with anti-FLAG antibodies (Sigma Aldrich, Milano, Italia), 1:10,000) and an HRP-conjugated anti-mouse antibody (Sigma Aldrich, Milano, Italia 1:100,000), followed by the enhanced chemiluminescence reaction (GE-Healthcare, Chicago, Illinois, USA) as described [41]. Images were acquired by FluorChemTM (Alpha Innotech) as selected areas of each membrane.

### 4.6. Fluorescence Microscopy

*Poly-L-lysine—coating of coverslips.* Coverslips were washed in 70% ethanol, rinsed in ddH_2_O and placed in petri dishes where they were submerged with a poly-L-lysine solution diluted 1:10 (Sigma Aldrich, Mialno, Italia), for 30 min at room temperature. The coverslips were rinsed with ddH_2_O and allowed to dry under a hood.

*Imaging.* Cultures of PAO1 and STM were pre-inoculated at 37 °C in LB broth overnight, diluted 1:500 in 1 mL of M9—S and VBMM, respectively, and grown for 5 h at 37 °C with shaking. The grown samples were then incubated for 20 min under the same conditions with compound **1** (0.05 mM). After washing twice sterile PBS, bacteria were resuspended in 0.5 mL PBS with 5 µL of Hoechst 33,342 (20 mg L^−1^) (Thermo Fisher Scientific, Waltham, MA, USA). After a 20 min incubation at room temperature with constant gently shaking, cells were centrifuged at 10 krpm for 1 min and then resuspended in 0.1 mL of fresh PBS. A total of 3 µL of each sample was spotted on a slide, covered with the poly-lysine coverslips and analyzed by live fluorescence microscopy using a Laica DMR fluorescence microscope equipped with a 100X objective and by LAS X software. Images were taken with CCD Microscope Camera Leica DFC3000 G.

## Figures and Tables

**Figure 1 ijms-22-10217-f001:**
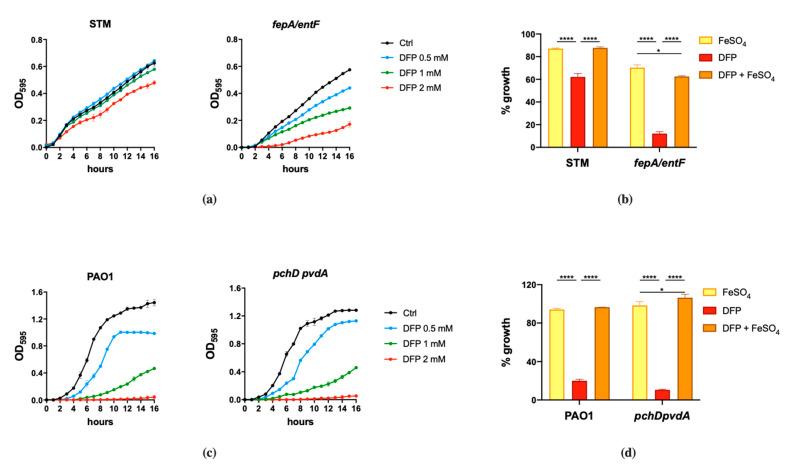
Effect of DFP on STM and PAO1. STM (wild type) and *fepA/entF* strains (**a**) and PAO1 (wild type) and *pchD pvdA* strains (**c**) were grown in LB medium with increasing amounts of DFP as indicated. In (**b**,**d**), the same strains were grown in presence of FeSO_4_ (0.5 mM), DFP (2 mM in (**b**) and 1 mM in (**d**)) or both, optical densities were measured after 14 h and % of growth were calculated taking the growth of untreated strains as 100%. Statistical significance (two-way ANOVA and Sidak’s multiple comparisons test): * *p* < 0.05; **** *p* < 0.0001.

**Figure 2 ijms-22-10217-f002:**
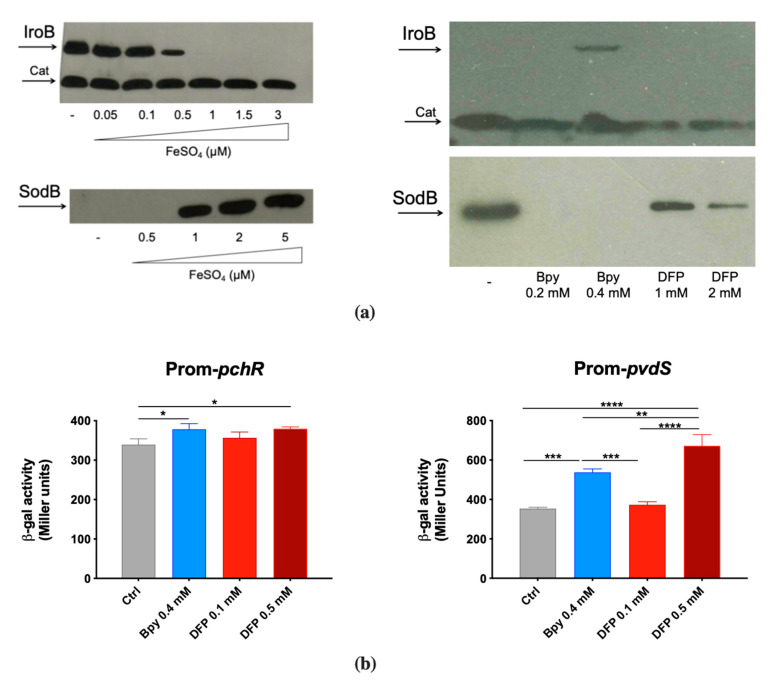
Responsiveness of Fur-regulated elements to DFP. (**a**) Western Blots of IroB and SodB proteins of STM from strains SA213 and MC120, respectively. Bacteria were grown for 18 h in M9 Minimal Medium supplemented with FeSO_4_ (**left**) or in LB medium supplemented with Bpy or DFP (**right**) as indicated. In lysates from strain SA213 the Cat protein is taken as an internal loading control. (**b**) Transcriptional activity of *pchR* and *pvdS* promoters carried on the pMP220 reporter plasmid in PAO1. Bacteria were grown in LB medium supplemented with Bpy or DFP as indicated. Statistical significance (one-way ANOVA and Tukey’s multiple comparisons test): * *p* < 0.05; ** *p* < 0.01; *** *p* < 0.005 **** *p* < 0.0001.

**Figure 3 ijms-22-10217-f003:**
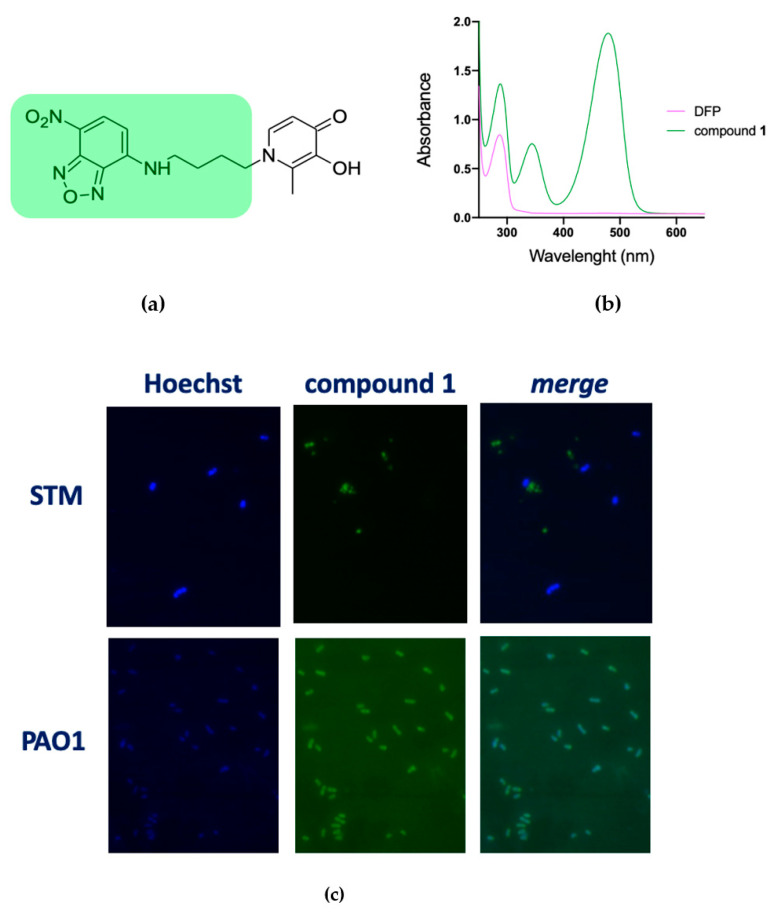
Fluorescence microscopy images of STM and PAO1 cells after treatment with compound 1. (**a**) Chemical structure of 1, highlighting the fluorescent moiety in green. (**b**) Representative absorbance spectrum of compound 1 (green line) compared to DFP (pink line). (**c**) Cells were processed and analyzed by fluorescence microscopy as described in the Materials and Methods. The experiments were done independently twice, and the best images are shown here.

**Figure 4 ijms-22-10217-f004:**
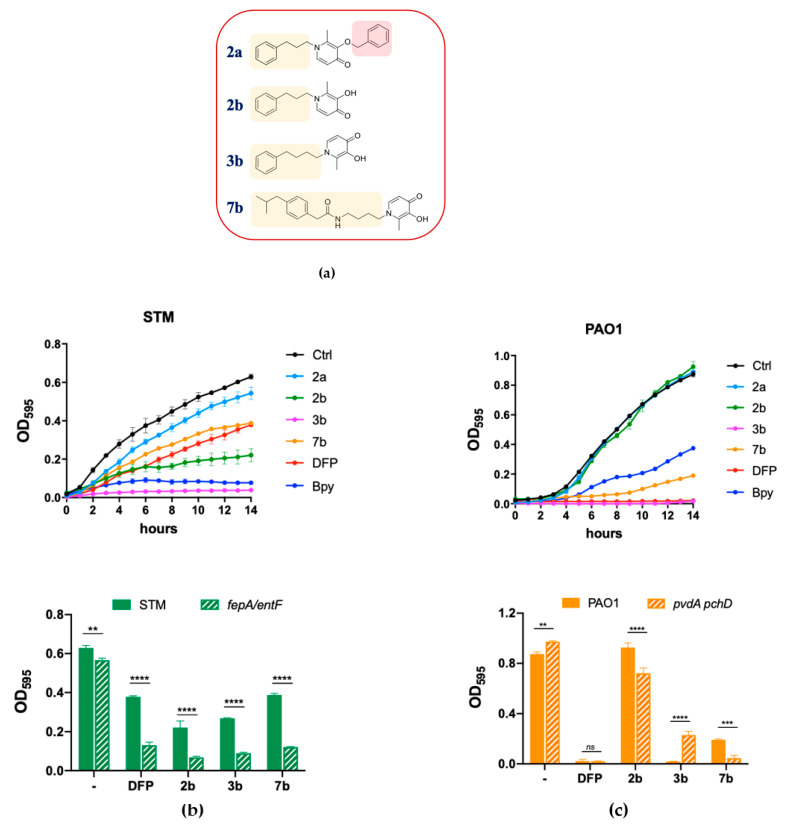
Effects of DFP derivatives on bacterial growth. (**a**) Molecular structures of the compounds selected for this study; the chemical moieties added to DFP are highlighted. (**b**) Effect on STM growth. In the upper graph the growth curves of STM in LB (Ctrl) supplemented with compounds 2a, 2b, 3b, 7b, DFP (2 mM) or Bpy (0.2 mM) are shown; in the lower graph the optical densities of STM (wild type) and *fepA/entF* mutant, after 14 h of growth in LB (−) supplemented with DFP, 2b, 3b or 7b (2 mM), are compared. (**c**) Effect on PAO1 growth. In the upper graph the growth curves of PAO1 in LB (Ctrl) supplemented with compounds 2a, 2b, 3b, 7b, DFP (2 mM) or Bpy (0.2 mM) are shown; in the lower graph the optical densities of PAO1 (wild type) and *pchD pvdA* mutant, after 14 h of growth in LB (-) supplemented with DFP, 2b, 3b or 7b (2 mM), are compared. Statistical significance (two-way ANOVA and Sidak’s multiple comparisons test): ** *p* < 0.01; *** *p* < 0.005; **** *p* < 0.0001, *ns*—non-significant.

**Figure 5 ijms-22-10217-f005:**
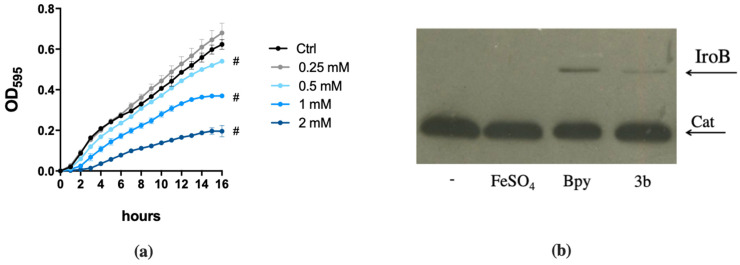
Effects of compound 3b on STM. (**a**) Dose-response of STM grown in LB medium (ctrl) with different amounts of compound 3b as indicated in the legend. Statistically significant differences between ctrl and treatments at the end point of the curves are indicated (two-way ANOVA and Tukey’s multiple comparison test; # *p* < 0.005). (**b**) Western Blots of IroB protein from strain SA213, grown for 18 h in LB supplemented with FeSO_4_ (0.2 mM), Bpy (0.1 mM) or compound 3b (1 mM). Cat protein is indicated as internal loading control. (**c**) and (**d**) STM wild type and *fepA/entF* strains were grown in (**c**) LB supplemented with FeSO_4_ (0.5 mM), compound **3b** (1 mM) or both and in (**d**) M9 minimal medium supplemented with DFP (0.2 mM), compound 3b (0.2 mM) or Bpy (0.2 mM). Optical densities were measured after 14 h and % of growth was calculated taking the growth of untreated strains as 100%. Statistical significance (two-way ANOVA and Sidak’s multiple comparisons test): * *p* < 0.05; ** *p* < 0.01; *** *p* < 0.005; **** *p* < 0.0001.

**Figure 6 ijms-22-10217-f006:**
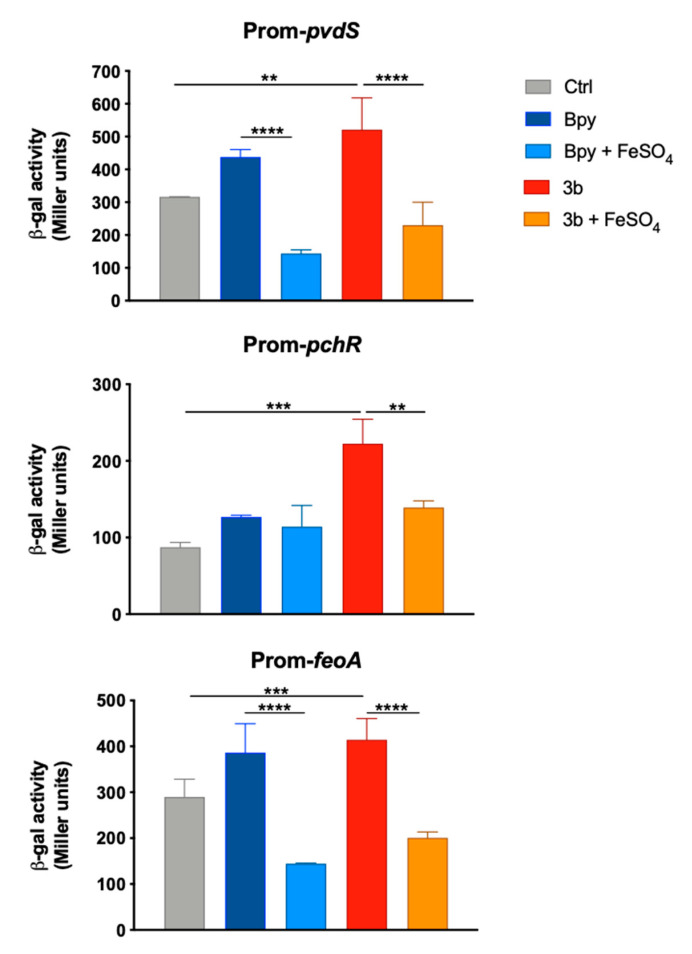
Transcriptional activity of *pvdS*, *pchR* and *feoA* promoters in PAO1. Wild type bacteria transformed with the pMP220 reporter carrying the promoter region of *pvdS*, *pchR* and *feoA* were grown in LB medium (Ctrl) supplemented with Bpy (0.2 mM) or compound 3b (0.2 mM), with or without FeSO_4_ (0.2 mM) as indicated. Statistical significance (one-way ANOVA and Tukey’s multiple comparisons test): ** *p* < 0.01; *** *p* < 0.005 **** *p* < 0.0001.

## Data Availability

Not applicable.

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
