# Peer review of "Salmonella* Typhimurium and *Pseudomonas aeruginosa* Respond Differently to the Fe Chelator Deferiprone and to Some Novel Deferiprone Derivatives"

_ijms, 2021, doi:10.3390/ijms221910217_

Round 1

Reviewer 1 Report

In their manuscript “Salmonella Typhimurium and Pseudomonas aeruginosa respond differently to the Fe chelator deferiprone and to some 3 novel deferiprone derivatives”, Ammendola et al.  describe the respond of two bacteria Salmonella enterica sv Typhimurium (STM) and Pseudomonas aeruginosa (PAO1) to deferiprone (DFP). DFP effectively inhibits the growth of PAO1, but not of STM. They also tested analogues of DFP and showed that some have a greater ability to interfere with Fe homeostasis in STM compared to DFP, whereas most, although not all, were less active than DFP against PAO1. the article is well written and the results presented clearly. However, I have different questions on the strategy (s) and on the interpretation of certain results:

Figure 1: why has this study been carried out in LB medium and not in an iron starved medium. During infection, bacteria are under iron starved conditions. It would have been more significant and closer to the reality to carry also these assays in an iron restricted medium. Is there production of pyoverdine in the growth conditions used here? According to the literature not really. It is very easy to follow the production of pyoverdine in PAO1 culture since the culture becomes green (specific absorbance of pyoverdine at 400 nm that can be followed during bacterial growth even in LB).

Figure 2B: I am not convinced by the effect of DFP on pchR. It would have been interesting to carry out this experiment under iron restricted condition where pyochelin and pyoverdine is produced. In LB very low amounts of these siderophores are produced according to the literature. Since pyochelin is produced more at the beginning of bacterial growth and pyoverdine latter in the culture, it would have been interesting to also carry out this assay at 8 h cultures for example.

Figure 3: show a zoom on a few fluorescent PAO1 bacteria in order to see if the fluorescence is homogenous all over the bacteria. What is the proof that the fluorescence seen by fluorescence microcopy is inside the bacteria and not bound at the cell surface. I am wondering if a cell fractionation in order to isolate the cytoplasm and periplasmic fractions wouldn’t have been necessary to proof that the fluorescence is inside the bacteria and not at the cell surface.

Concerning the conclusion, an uptake or diffusion of the fluorescent probe does absolutely not mean that DFP goes inside the bacteria, because these are two different molecules with different chemical properties and may be also different biological properties. This fluorescent probe should have been tested in the assays presented in figures 1 and 2 to show that it behaves like DFP on P. aeruginosa.

At last, it is important to determine and present MIC values for the tested molecules (Table S2) to have a better idea of their ability to affect bacterial growth on PAO1 and STM. This is a key point missing.

Minor comments:

Line 37: I am not sure that I understand the meaning of this sentence. Getting access to iron is a trivial problem for bacteria despite its abundance on earth.

Line 77-78: give references, because there are much more articles proposing vectorization of antibiotics by siderophores than iron chelating strategies.

Line 118 and 120, precise somewhere that pyochelin and pyoverdine are the two siderophores produced by P. aeruginosa and enterobactin is produced by STM. It is mentioned nowhere in the introduction or in the results.

Line 174: this sentence should be modified: “This compound has an absorption peak at 480 nm (Figure 3b) and therefore it could be imaged under a fluorescence microscope using the FITC excitation filter.” It is not because a molecule has a peak of absorbance that it is fluorescent.

Line 177: is it a short growth of 20 min or just an incubation?

Table S2 should be moved in the main text, important to show the molecules. I do not understand figure 4a.

Line 226: a punctuation is missing after “strain”.

Line 324: ka = 1042 M-1, is the affinity of DFP for iron given in reference 28, if yes, give the value in the discussion. Please give also the affinities of pyochelin and pyoverdine for iron, 1018 M-2 and 1032 M-1 respectively.

Reviewer 2 Report

This manuscript compares the reaction of S. Typhimurium and P. aeruginosa to deferiprone and some of its derivatives. As Fe chelators are important in the pathogenicity of bacteria, the presented research is necessary.

Generally, the manuscript is very well written, however, I have several remarks for the authors to consider before the paper can be accepted for publication. 

  1. The keywords repeat the names of bacteria that are already present in the title. This repetition is not necessary. 
  2. Line 100 - a part of the word enterica is written without italics. 
  3. ATCC strains should be written with all due trademarks as indicated in the strain infosheet.
  4. Was P. aeruginosa PAO1 a 'fresh' strain from ATCC collection or rather a strain with a history of passaging? It was shown that strains kept in various labs can be quite different.
  5. Can you provide full images of Western blots used in Figs. 2 and 5? Please provide them in the supplementary material. Furthermore, the quality of bands in upper left panel (cat) does not look good.
  6. Similar to the previous comment. Can you provide more figures from epifluorescence in the supplementary material. It would be great to show your readers from which images you have selected the ones that are presented in the main manuscript. Otherwise, you may be suspected of cherry-picking. 
  7. References: around 1/3 of references are no older than five years. It is not a fault to use older references, however, it would be good for the sake of the study's novelty to refresh the list if possible. The selected references look very good. I did not detect any citation from a suspected predatory journal.
  8. I suggest introducing a separate conclusion section in the manuscript.

Round 2

Reviewer 1 Report

There are still two errors. The affinity of pyochelin for iron is 1018 M-2  (and not -1) because of the stoechiometry of chelation of 2 pyochelins for one iron ion.

the reference of pyoverdien affinity for iron is Albracht-Gary norg. Chem. 1994, 33, 26, 6391–6402, and not ref 33 given by the authors.

Author Response

We really thank the reviewer for the careful check.

We have now corrected the value for the affinity constant of pyochelin and inserted the original reference reporting the determination of pyoverdine affinity for iron

Reviewer 2 Report

Dear Authors, 

thank you for submitting the revised version of your manuscript.

All my suggestions have been taken into consideration. I do not have any additional remarks and recommend this manuscript for publication. 

Author Response

We wish to thank the reviewer for the fast and positive evaluation of the revised manuscript